# Impact of Hydrophobic and Electrostatic Forces on the Adsorption of *Acacia* Gum on Oxide Surfaces Revealed by QCM-D

Athénaïs Davantès [1],*, Michaël Nigen [2], Christian Sanchez [2] and Denis Renard [1]

[1]  INRAE, UR BIA, 44316 Nantes, France
[2]  UMR IATE, Université Montpellier, INRAE, Institut Agro, 34060 Montpellier, France
*   Correspondence: athenais.davantes@inrae.fr

**Abstract:** The adsorption of *Acacia* gum from two plant exudates, *A. senegal* and *A. seyal*, at the solid-liquid interface on oxide surfaces was studied using a quartz crystal microbalance with dissipation monitoring (QCM-D). The impact of the hydrophobic and electrostatic forces on the adsorption capacity was investigated by different surface, hydrophobicity, and charge properties, and by varying the ionic strength or the pH. The results highlight that hydrophobic forces have higher impacts than electrostatic forces on the *Acacia* gum adsorption on the oxide surface. The *Acacia* gum adsorption capacity is higher on hydrophobic surfaces compared to hydrophilic ones and presents a higher stability with negatively charged surfaces. The structural configuration and charge of *Acacia* gum in the first part of the adsorption process are important parameters. *Acacia* gum displays an extraordinary ability to adapt to surface properties through rearrangements, conformational changes, and/or dehydration processes in order to reach the steadiest state on the solid surface. Rheological analysis from QCM-D data shows that the *A. senegal* layers present a viscous behavior on the hydrophilic surface and a viscoelastic behavior on more hydrophobic ones. On the contrary, *A. seyal* layers show elastic behavior on all surfaces according to the Voigt model or a viscous behavior on the hydrophobic surface when considering the power-law model.

**Keywords:** adsorption; oxide; arabinogalactan-protein; viscoelastic; rheology; interface





## 1. Introduction

*Acacia* gum, or gum Arabic, is the exudate from the *Acacia senegal* (*A. senegal*) and *Acacia seyal* (*A. seyal*) trees, which is widely used in the industry in many fields of application since the dawn of time. Its remarkable capacity as an emulsifier, stabilizer, thickener, encapsulating agent, or flavoring agent makes it an essential part of food formulations ranging from ice creams, candies, soft drinks, wines, jellies, beverages, syrups, or chewing gums [1,2]. It is also used in cosmetics for the formulation of creams and lotions. In the pharmaceutical industry, *Acacia* gum is used in drug delivery or as an anti-inflammatory agent and in sensor and tumor imaging [3]. Its binding properties makes gum a key ingredient in lithography and also widely used in the textile industry. Now, regarding organic and green chemistry, many studies focus on the potential of the use of gum in nanotechnology to reduce the nanoparticles' toxicity and increase their stability [4–7]. The use of *Acacia* gum is increasingly implicated in the medical field, especially in nano-medicines and biosensors [3]. Studies show the incredible potential of *Acacia* gum in the nano-medicines as a potential treatment for long tumors combined with gold nanoparticles [8], in anti-tumor methods against breast, pancreatic and prostate cancers [9], in effective therapies for mistreated melanomas with gold-nanorod composites [10,11], or as biodegradable antibacterial material [12]. The growing interest in the combined use of *Acacia* gum and oxides makes it important to better understand the adsorption mechanisms and its behavior on these surfaces.

*Acacia* gum is a complex heterogeneous material composed of a continuum of hyperbranched arabinogalactan-proteins (AGPs) differing by their protein to sugar ratio, molecular weight, and charges [2,13]. AGPs macromolecules represent about 94–96% of the total compounds in *Acacia senegal* and *Acacia seyal* gums. The remaining percentages are mainly minerals [2]. Generally, *A. senegal* gum is richer in proteins than *A. seyal* (2.0 wt% vs. 1.0 wt%, respectively), and more polydisperse with the highest proportion of high molecular weight AGPs [14,15]. However, AGPs from *A. seyal* have a different structure and a more compact conformation than *A. senegal* ones, with conformations varying from spheres to oblate ellipsoids, while AGPs from *A. senegal* vary from oblate ellipsoids to more anisotropic conformations [15,16].

Previous studies highlighted some important differences in the adsorption capacity and behavior of both gums on gold surfaces, with the highest adsorption capacity being for *A. senegal* gum. This study showed that hydrophobic forces would drive the adsorption process between the polypeptide backbone and the surface, while the swelling capacity would depend on the hydrophobic and hydrophilic effects between polysaccharide blocks and water [17]. Our recent study demonstrated that electrostatic interactions also played an important role in AGPs' adsorption, and highlighted that the adsorption process was mostly driven by both the protein moiety and its molecular weight distribution, with the structural accessibility as a key factor [18].

Hydrophobic force, also known as the hydrophobic effect or hydrophobic interaction, is a phenomenon that refers to the unusually strong interaction between hydrophobic molecules and surfaces in water that cause hydrophobic moieties to aggregate or cluster [19–22]. Yet, it is commonly accepted that the physical driving force underlying hydrophobic interactions is that water molecules cannot form hydrogen bonding with hydrophobic moieties leading to specific orientations, adversely affecting the hydrogen bonding network and losing configurational entropy [23]. Therefore, reducing the contact between water molecules and apolar surfaces increases the water configurational entropy, resulting in an attraction between two non-polar objects in sufficient proximity that can be identified as hydrophobic forces or interactions [21]. Although this interpretation is not ideal and can be discussed, many studies cover this fascinating topic by attempting to measure and rationalize the mechanism involved in hydrophobic forces [20–26].

According to our previous studies [17,18], both hydrophobic and electrostatic forces are important during *Acacia* gum adsorption. However, several questions remain unanswered. Is there a limiting factor for the adsorption capacity between hydrophobic and electrostatic forces? Which parameters rule *Acacia* gum adsorption? Can we predict gum adsorption only by controlling the surface properties? In order to answer these questions, the adsorption of the two main gums (*A. senegal* and *A. seyal*) was followed by a quartz crystal microbalance with dissipation monitoring methods (QCM-D) on various oxide surfaces with different properties. The use of this technique allows for the simultaneous measurement of the mass and the thickness. QCM-D also allows for the determination of the viscoelastic and rheological properties of the film adsorbed on the surface. The impacts of the surface properties such as hydrophobicity, charge, and roughness were investigated through their surface oxide nature (i.e., gold, silicon dioxide, and titanium dioxide), and the variation of the salt concentration or pH solutions.

## 2. Materials and Methods

### 2.1. Materials

The experiments were carried out using commercially available *Acacia senegal* (*A. senegal*, lot OF152413) and *Acacia seyal* (*A. seyal*, lot OF110724) soluble powders, provided by the Alland and Robert Company—Natural and Organic gums (Port Mort, France). A summary of the biochemical composition and structural parameters of both gums used in this study is provided in the supporting information (Table S1).

Dilutions of the stock gum dispersions (C = 10 g/L) to the desired concentrations were performed in 10 mM acetate (VWR Chemicals, Fontenay-sous-Bois, France) containing or

not containing NaCl salt. The gum concentrations in the stock dispersions were quantified by the dry matter method.

The impact of ionic concentrations was studied by adding NaCl in the solution at pH 5.0, between 0 mM, corresponding to the solution containing only 10 mM acetate, and 100 mM NaCl. The pH varied from 3 to 8 using the HCl or NaOH solution (Merck, analytical grade, Molsheim, France). All stock solutions and dispersions were prepared at room temperature and filtered with a 0.2 μm filter unit (GHP, Pall Laboratory, Life Sciences, St. Germain-en-Laye, France) using fresh purified water (Milli-Q, Millipore, Staffordshire, UK) with a resistivity of 18.2 MΩ·cm.

### 2.2. Quartz Crystal Microbalance with Dissipation Monitoring (QCM-D)

The adsorption of *Acacia* gum was carried out using a Q-Sense E4 instrument (Gothenburg, Sweden), using a piezoelectric AT-cut quartz crystal coated with gold electrodes, with a nominal resonance frequency of 5 MHz. Briefly, this method allows us to probe the mass change, the layer thickness, the reversibility of the adsorption process, the kinetics, and the process of macromolecule swelling by monitoring two parameters: the change in the resonance frequency ($\Delta F$) and the change in the dissipation energy ($\Delta D$) [17,27,28]. The adsorption of molecules caused a decrease in the resonance frequency monitored as a function of time, while the dissipation energy was related to the viscoelastic properties of the molecular layers.

Different QCM-D crystals were purchased from Q-sense (Gothenburg, Sweden): gold-coated quartz crystal (QSX 301), silicon dioxide-coated quartz crystal (QSX 303), and titanium dioxide-coated quartz crystal (QSX 900). Prior to use, all surfaces were treated with UV/Ozone (Jelight Company Inc., Irvine, CA, USA) for 10 min ($SiO_2$ and $TiO_2$) or 15 min (Gold) to remove hydrocarbon and organic contaminants and render the surface hydrophilic. After QCM-D experiments, the quartz crystals were cleaned according to different protocols for reuse. All quartz crystals were immersed in the SDS 2% solution for 30 min and rinsed with Milli-Q water. The gold-coated quartz crystals were then cleaned in piranha solution $H_2SO_4/H_2O_2$ (7:3, *v/v*) for 3 min, rinsed exhaustively with Milli-Q water, and dried under a stream of nitrogen. The $TiO_2$ quartz crystals were sonicated in ethanol for 10 min, rinsed with Milli-Q water, and dried with nitrogen stream. The $SiO_2$ quartz crystals were immersed in isopropanol solution for 10 min, rinsed with Milli-Q water, and dried with nitrogen stream. $SiO_2$ quartz were then heated at 100 °C for one hour, and sonicated in ethanol for 10 min, rinsed with Milli-Q water, and dried with nitrogen stream.

All experiments were carried out at 20 °C with a flow rate of 200 μL/min in an independent closed-system configuration for each cell. At least one hour of stabilization with the solution at the specific experimental conditions (ionic strength and pH) was needed to reach the equilibration state corresponding to the baseline conditions. *Acacia* gum concentrations were chosen according to the maximum coverage concentration probed by isotherm [17], i.e., 150 ppm for *A. senegal* and 500 ppm for *A. seyal*. The gum solution was left in contact with the substrate until stabilization was reached during the adsorption process. Desorption was investigated by rinsing the system with the initial buffer solution.

Interpretations of QCM-D data are tied to the model used to describe the viscoelastic properties of the film [29–34]. The Sauerbrey equation is used to extract the area mass $\Gamma_{QCM-D}$ and the thickness $d_{QCM-D}$ of the film when the dissipation changes are very low ($\Delta D < 1 \times 10^{-6}$) and frequency overtones are homogeneous:

$$\Gamma_{QCM-D} = -C\frac{\Delta F_n}{n} = d_{QCM-D} \times \rho \tag{1}$$

where $C$ is the mass sensitivity constant of the quartz crystal (17.7 ng·cm$^{-2}$), $n$ is the overtone number, and $\rho$ is the film density. The Sauerbrey equation is only valid for a homogeneous and rigid film. This model was used only for one condition in this study (i.e., *A. senegal* on $SiO_2$ at 0 mM NaCl), where the adsorption was too low to apply the viscoelastic model.

A commonly used model for viscoelastic film, when $\Delta D > 1 \times 10^{-6}$ and the overtones are inhomogeneous, is the Voigt model [29,30,34]. In this model, the adsorbed film is represented by a complex shear modulus $G$, defined as:

$$G = G' + iG'' = \mu_f + 2\pi f \eta_f \tag{2}$$

where $\mu_f$ is the film shear elastic modulus, $f$ is the oscillation frequency, $\eta_f$ is the film shear viscosity, $G'$ is the apparent film storage modulus, and $G''$ is the apparent film loss modulus. Another more realistic power-law model can be used in some cases, with exponents $\alpha'$ and $\alpha''$: $G'(\omega) = G\prime_{ref}(\frac{\omega}{\omega_{ref}})^{\alpha'}$ and $G''(\omega) = G''_{ref}(\frac{\omega}{\omega_{ref}})^{\alpha''}$, where $\omega$ is the angular frequency of deformation [30–33]. The power-law model gives a higher fitting quality and reliable results; however, in our calculations, this model fails when the film was not "viscoelastic" enough, i.e., $\Delta D \geq 3 \times 10^{-6}$ and high inhomogeneity of overtones were needed to give reliable results.

The adsorption mass, film thickness, shear elastic modulus, and shear viscosity were calculated using both the Voigt and power-law modelling with the Dfind software with at least five overtones with a good signal/noise ratio (Biolin Scientific, Q-Sense, Gothenburg, Sweden). The validity of a model was chosen according to the fit quality, $\chi^2$, and if all general trends were physically reasonable compared to each experimental conditions used and reliable with our previous studies. The adsorbed film was assumed to have both a uniform thickness and density. The film density was assumed to be the inverse of the partial specific volume $v_s°$, $\rho = \frac{1}{\mathbf{vs°}}$ (see data in Table S1) [35,36], i.e., 1.703 cm$^3$·g$^{-1}$ for *A. senegal* and 1.734 cm$^3$·g$^{-1}$ for *A. seyal*, respectively. All experiments were conducted at least three times with both new and reused quartz, and all calculated data presented in this study corresponded to the mean ± standard deviation.

### 2.3. Surface Properties

Static contact angle measurements were performed with a Tracker$^{TM}$ automatic drop tensiometer (Teclis Scientific, Civrieux d'Azergues, France) in the method of sessile down. Images were analyzed with the WDROP software (Teclis Scientific, Civrieux d'Azergues, France). A drop of 5 μL of a solution containing 10 mM acetate and 50 mM NaCl at the desired pH was deposited on the cleaned sensor surface at room temperature. Images were recorded for 60 s and the measurements obtained at a stable state (i.e., 30 s) are presented in this study. At least five measurements on different surfaces were averaged to obtain reliable results.

Roughness images of bare surfaces were measured by atomic force microscopy (AFM) in the air using INNOVA (Bruker, Palaiseau, France). The images were recorded with low scan rates between 0.5 and 1 Hz, using FESPA-V2 tips (Bruker) with 2.8 N m$^{-1}$ spring constants in the tapping mode. The root-mean-square roughness (RMS) and the roughness factor $r$ were extracted from three separate images obtained in different regions on each cleaned surface with the Gwyddion software (freeware available at http://gwyddion.net/, accessed on 9 February 2021). $r$ is the ratio between the real 3D surface area ($A$) of the topography determined by AFM to the projected 2D flat surface area ($A_g$) of the topography, defined as [37,38]:

$$r = roughness\ factor = \frac{A}{A_g} \tag{3}$$

## 3. Results and Discussion

### 3.1. Characterization of Quartz Sensors

Electrostatic, hydrophobic, and hydrophilic interactions or forces are generally known as important driving forces for the adsorption of proteins on solid-liquid interfaces [39–43]. Therefore, the solid surfaces used in this work were chosen to probe these interactions using different surface properties.

The degree of hydrophobicity of a solid surface is generally defined by its contact angle with water in the air, where values above 90° are considered hydrophobic and values below 90° are considered hydrophilic [44]. The surface hydrophobicity was probed using the static contact angle at different pHs (Figure 1). The SiO$_2$ sensor is a very hydrophilic surface with an 11° contact angle, while the gold sensor is a more hydrophobic surface (i.e., 77°). Both surfaces present a constant behavior over the entire pH range. However, the TiO$_2$ surfaces present a different behavior. The contact angle behaviors of the TiO$_2$ sensor and especially the relaxation mechanism are not yet fully understood phenomena, even if they are, nevertheless, well studied [45–47]. It is well known that the TiO$_2$ surface presents a hydrophilic behavior, which switches to hydrophobic after UV exposure, but the same phenomenon is observed in the dark over time. Some models have suggested that carbon contamination, and the loss of surface OH generated by UV light, may be the major causes. This phenomenon called relaxation behavior was observed in this study by comparing new sensors with sensors that were previously used (Figure 1). Indeed, the contact angle increases after several uses of the sensors from hydrophilic (20–40°) to relatively less hydrophilic (40–60°), but always presents an average hydrophilic behavior between the two other sensors (gold and silica). Indeed, both the exposure to UV light and carbon contamination can be responsible for the change of surface contact angle in this study. In our QCM-D experiments, no impact of this hydrophilicity change was observed and the results were reproducible. Indeed, all sensors were UV/Ozone cleaned before use, which may sufficiently correct this relaxation behavior during the QCM-D experiments. This contact angle variation was only observed for the TiO$_2$ sensors.

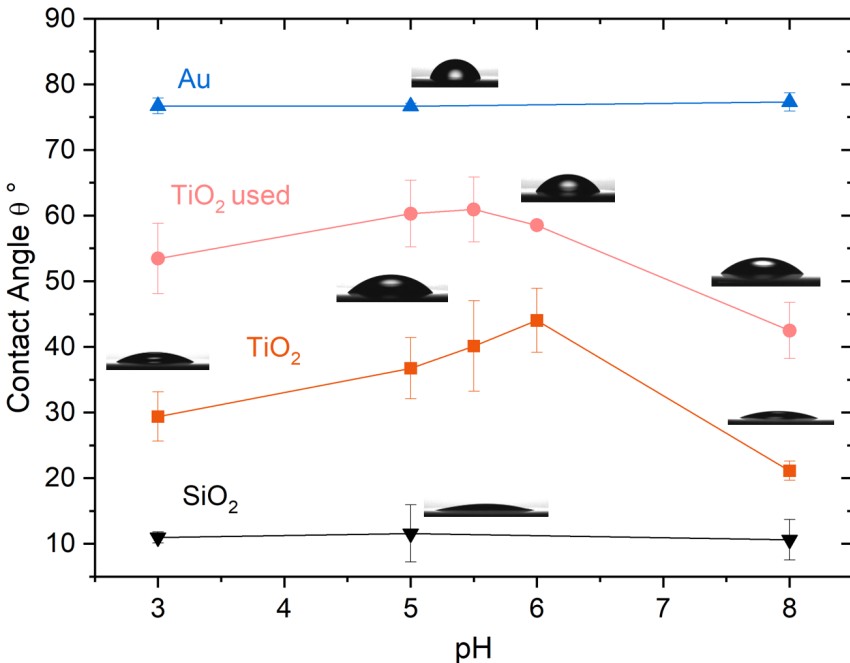

**Figure 1.** Sensors surface contact angle in function of pH solutions: Au (-▲-); TiO$_2$ (-■-); TiO$_2$ sensors previously used (-●-) and SiO$_2$ (-▼-).

The charge surface is an important factor to probe the electrostatic interactions. The isoelectric point (pH$_{iep}$) of a gold surface is around 2.9 [48,49], while the silica surface presents a pH$_{iep}$ between 2 and 3 [50–53]. Therefore, these two types of sensors present a negative charge surface across the pH range studied. The TiO$_2$ sensor is an amphoteric surface that can be both positively and negatively charged, depending on the pH in the solution relative to its isoelectric point. In the literature, most of studies found a pH$_{iep}$ for TiO$_2$ around 5.8 [50,53–58]. Further studies used the contact angle titration as a function of pH to determine the isoelectric point of metal [59] and metal oxide [60–63], such as amphoteric surfaces [64,65]. The advancing contact angle is at the maximum when

approaching the isoelectric point of the surface. As observed in Figure 1, there is an increase in the contact angle of TiO$_2$ surfaces between pH$_{iep}$ 5.5 and 6, in agreement with data in the literature. Therefore, the TiO$_2$ sensors can be either positively (pH < 5.5) or negatively (pH > 6.0) charged, depending on the experimental pH conditions. No significant contact angle changes were observed for the gold or silica surfaces within the pH range, which validated pH$_{iep}$ < 3 for both surfaces. Interestingly, Virga et al. [65] noticed that an increase of ionic strength leads to an increase in hydrophilicity, away from pH$_{iep}$, on the amphoteric surface. However, van Oss [64] showed that there was no significant influence on the surface properties in aqueous solutions with low salt concentrations (<0.2 M). In this study, no significant change on the surface polarity was observed on all sensor surfaces at pH 5.0 by varying the ionic strength (data not shown).

The roughness of the sensor surfaces was investigated through AFM characterization with the determination of the RMS and the roughness factor *r*, as displayed in Table 1. AFM representative topologic images of the three types of quartz sensors used in this study are shown in the supporting information (Figure S1). Generally, the TiO$_2$ sensors present the smoother surfaces while the gold surfaces are trougher, with $1.06 \pm 0.04$ nm and $1.76 \pm 0.08$ nm, respectively. No roughness variation was observed on the TiO$_2$ sensors before and after use. As previously observed [66,67], the sensors roughness can be an important factor for the adsorption that cannot be neglected. The main characteristics of the three sensors used in this study are summarized in Table 1.

**Table 1.** Quartz crystal sensors surface properties.

| Quartz | Au | SiO$_2$ | TiO$_2$ |
|---|---|---|---|
| Contact Angle θ $^\circ$ | $77.0 \pm 1.2$ | $11.1 \pm 3.1$ | 20–60 |
| Isoelectric Point | 2.9–3.4 | 2 | 5.5–6 |
| RMS (nm) | $1.76 \pm 0.08$ | $1.26 \pm 0.09$ | $1.06 \pm 0.04$ |
| *r* | $1.006 \pm 0.004$ | $1.006 \pm 0.005$ | $1.005 \pm 0.005$ |

*3.2. Acacia Gum Adsorption*

The adsorption of *Acacia* gum (*A. senegal* and *A. seyal*) on the three types of sensors was followed using the QCM-D technique, recording the $\Delta F$ and $\Delta D$ variations in time. Two types of experimental conditions were performed: fixed pH and varying ionic strength, by varying the salt concentration (NaCl), and varying the pH at constant ionic strength. The experimental curves are shown in the supporting information from Figures S2–S7. All frequency overtones and dissipation energies ($\Delta D$) were highly inhomogeneous. which is characteristic of an adsorbed viscoelastic film (Figure S8 shows the $\Delta F$ and $\Delta D$ splitting of *A. senegal* at pH 5.0 and 50 mM NaCl for illustration). The power-law model was successfully used for all *A. senegal* results. However, this enhanced model was not appropriate for all *A. seyal* data. Both viscoelastic models follow the same trend (Figure S9); thus, for data comparison, the Voigt model was used on all *A. seyal* results. Detailed results of the model calculations are presented in Tables S2–S13.

The comparison of the adsorbed mass of *A. senegal* and *A. seyal* on each surface in function of the ionic strength at pH 5.0 is presented in Figure 2a,b, respectively. As observed previously [17,18], despite the electrostatic repulsions between both negatively charged gold surfaces and *Acacia* gums, a significant adsorption was observed (blue lines). However, contrary to what should be expected, the adsorption on a positively charged surface as TiO$_2$ (orange lines) was lower than on the negatively gold-charged surface. The silicon dioxide sensors presented a lower adsorption capacity. Nevertheless, it can be concluded in the first approximation that the adsorption process seemed to follow the surface hydrophobicity for both gums.

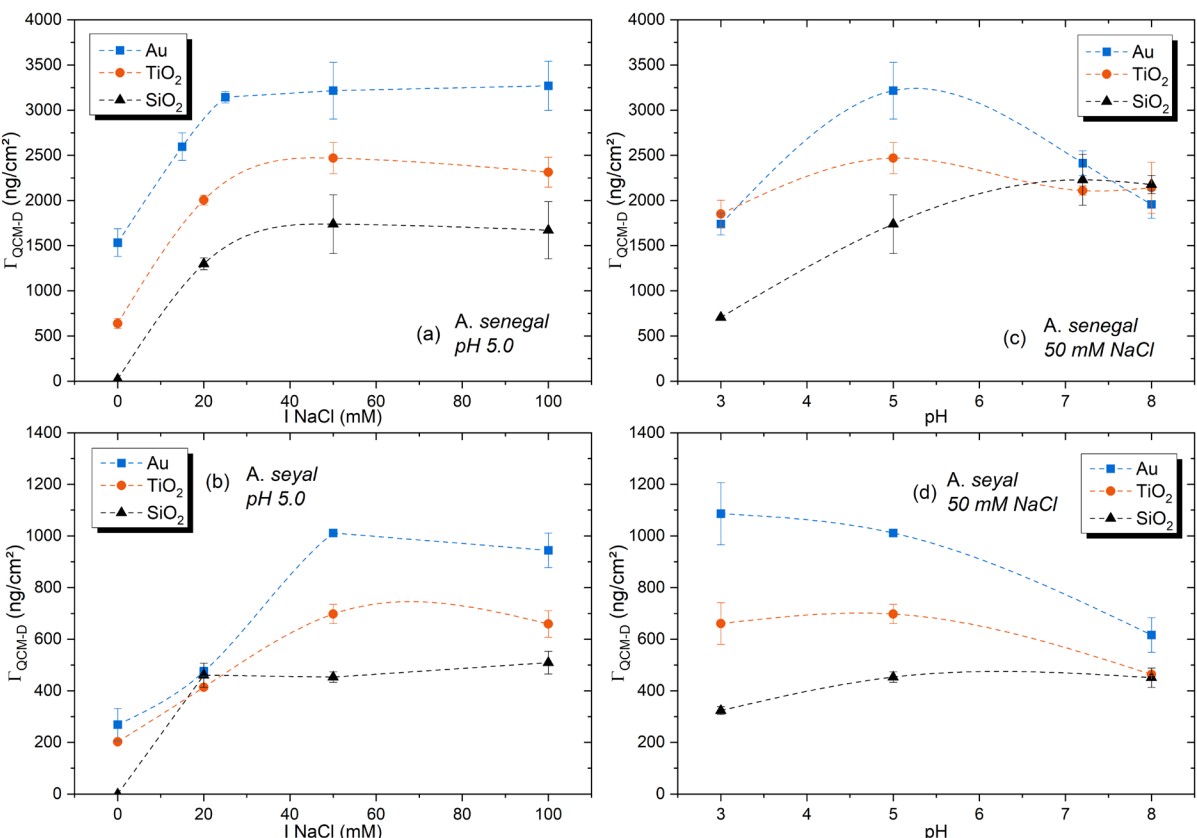

**Figure 2.** Adsorbed amount ($\Gamma_{QCM-D}$) onto Au (-■-), TiO$_2$ (-●-) and SiO$_2$ (-▲-) substrates at equilibrium state in function of NaCl ionic strength (**a**,**b**) at pH 5.0, and in function of pH at 50 mM NaCl (**c**,**d**) for *A. senegal* and *A. seyal* gums. *Lines are here to guide the eyes.*

    *A. seyal* gum presented a lower adsorption than *A. senegal* gum on all sensors. Indeed, *A. seyal* contains lower protein and HM$_w$ AGPs, and seems to be less apolar than *A. senegal* (Table S1), which are the three main parameters for adsorption efficiency [18]. Increasing the salt concentration leads to a screening of both the gum and surface charges, which reduced the intra- and inter-molecular electrostatic repulsions inside the gum macromolecules, but also the repulsion or attraction between the substrate and the gum. Adsorption can be, therefore, increasing, especially in the case of repulsive interactions such as gold or silica substrates. Despite the electrostatic attraction between *Acacia* gum and TiO$_2$ sensors, a mass increase was observed with the charge shielding (i.e., the decrease in electrostatic attraction). Furthermore, the same general trends were observed for both the gold and TiO$_2$ surfaces, while their respective surface charges were opposite, which could indicate that the electrostatic attraction had a limited influence on the gum adsorption. Interestingly, there is almost no adsorption for both gums on the silica surface if there is no salt compensation in the solution. This result showed that a sufficient change of polarity is necessary for the gum adsorption on the surface. At these experimental conditions, with a hydrophilic surface, the system energy is at its minimum and no adsorption is required to decrease it, which is not the case on the hydrophobic surface. This behavior is similar to the globular protein adsorption on hydrophilic surfaces, especially in charge-unfavorable conditions [68], with the result again highlighting the importance of the protein part in the *Acacia* gum adsorption. Stabilization was observed around 50 mM of NaCl in the solution for all surfaces, which could indicate that the maximum charges screening and/or the maximum adsorption capacity were reached. However, *A. seyal* presented a constant adsorption on the silica surfaces when the charge screening was enough to allow adsorption at 20 mM NaCl. Thus, on very hydrophilic surface, the variation of ionic strength had little impact on the *A. seyal* adsorption.

The comparisons of the adsorbed mass of *A. senegal* and *A. seyal* gum on each surface in function of the pH solution at 50 mM NaCl are presented in Figure 2c,d, respectively. Interestingly, while the gold and $TiO_2$ surfaces presented opposite charges between pH 3.0 and 5.0, the films' surface presented similar behavior. Moreover, despite the electrostatic attraction between negatively charged *Acacia* gum and positively charged $TiO_2$, the adsorption capacity was still lower or equivalent than on the gold surface. The surface charge, therefore, had little impact on the adsorption capacity compared to its hydrophobicity state. However, the number of charges on substrates may have been too low to have a real impact on the gum adsorption.

The acid dissociation constant ($pK_a$) of *Acacia* gum was determined by Grein-Iankovski et al. [69] with two characteristic $pK_a$ values: $pK_{a1}$ between 3 and 4 corresponding to the carboxylic group of the polysaccharide parts (uronic acid) and $pK_{a2}$ around 6.5 corresponding to the amine group of the protein moiety (Lys, Arg, His). A third $pK_a$ may be considered with the carboxylic groups of the protein backbone (Glu, Asp), with $pK_{a3}$ = 3.9–4.2. Generally, *Acacia* gum macromolecules adsorbed on a negative charged surface through protein moiety would carry positive charges, while on positive charged surface the polysaccharidic blocks would adsorb preferentially [17,70]. At pH 3.0, amine groups are protonated ($NH_3^+$), leading to an increase in the electrostatic attraction with the negatively charged surface, while the opposite phenomenon is observed between the positively charged surface ($TiO_2$) and the protein moiety of the gum. However, at a low pH, *Acacia* gum presented a very constrained conformation with "closed" protein structure, as suggested by Ma et al. [71], leading to a decrease in the adsorption capacity [17,71]. At a high pH, gum charges change to mostly negative, with carboxylate groups fully dissociated, and all surfaces are negative. The electrostatic repulsions are maximal, limiting the adsorption. At pH 5.0, *Acacia* gum presents both charges in its structure, with negative carboxylate groups and positive amine groups. Both attractive and repulsive electrostatic interactions are, therefore, presented regardless of the surface charge. However, the coexistence of both charge groups inside the gum structure could induce intra-molecular interactions between the polysaccharide and protein parts, which may lead to conformational constraints, as observed in the previous study [17]. The present study is, therefore, performed at a higher ionic strength (50 mM NaCl), which leads to a significant increase in the gum adsorption with charge shielding. The gum structure would be, therefore, more favorable to adsorption in these partly screened electrostatic conditions. Moreover, as previously observed, the hydration state of *Acacia* gum is maximized at pH 5.0 [17]. As the QCM-D technique records the "wet" mass adsorbed on the surface, including water involved in hydration shells and water molecules "trapped" inside the adsorbed layer, an increase in the adsorbed mass was observed. *A. seyal* contained less protein and charged groups than *A. senegal*, and was also less apolar, which reduces the adsorption at pH 5.0. Therefore, the charge state of the gum (i.e., $pK_a$) and its related structural conformation, is an important parameter of the adsorption process compared to the surface charge.

Interestingly, the more hydrophilic surface ($SiO_2$) presents a different behavior with a very slow adsorption process (Figures S6 and S7). Hydrophilic interactions are driven by the polysaccharide blocks of *Acacia* gums [17]. Although electrostatic repulsive interactions increase, the adsorbed mass slightly increases with pH, illustrating a different adsorption behavior on hydrophilic surface. Ma et al. [71] suggested that *Acacia* gum presents an expanded structure at pH 8.0 due to the increased repulsion of charged amino acids at high pH conditions. This "opening" structure allows the increase in hydrophilic interactions and the increase in the gum swelling capacity. The competition between water molecules on the strong hydrophilic surface and *Acacia* gums polysaccharide blocks may justify the slower and lower adsorption compared to more hydrophobic surfaces. However, one can notice that the adsorption reaches a similar capacity for the three surfaces at the higher pH condition. At a high pH, the surface degree of hydrophobicity, therefore, had a lower impact than the gum structural configuration.

A recent study reported a similar behavior on charged latex particles, with the highest adsorption for *A. senegal* on negatively charged latex particles [70]. However, the inverse was observed for *A. seyal*, with a maximum adsorption on positively charged latex particles. Moreover, a significant increase in the adsorption capacity was observed at pH 3 and 4. These discrepancies may arise from experimental conditions such as the adsorption kinetics (days vs. hours), the shape of the surface (3D surface vs. 2D flat surface), the number of surface charges (very high on latex particle vs. weak on oxide surface), and charge shielding with ionic strength, resulting in the very complex adsorption mechanism of *Acacia* gums depending on the experimental conditions.

Desorption was investigated by rinsing the system with the initial buffer solution after adsorption (Figure S10). *A. senegal* presented a very stable film on the negatively charged surface with less than 3% desorption, except when the adsorption was very low. Indeed, as previously observed [17], the film was more stabilized when the amount of gum was sufficient. However, despite a sufficient mass adsorption on the $TiO_2$ sensors, *A. senegal* layer presented less stability on this surface. Interestingly, the lower layer stability seemed to be correlated with the surface charge, with a maximum desorption process at pH 3.0, with 20.3% of film desorption, while the film was stabilized at pH 8.0, with 1.8% of film desorption when the surface became negatively charged. This phenomenon was also observed with ionic strength, with a better stabilization when the charge shielding became sufficient (50 mM). On a positively charged surface, *A. senegal* layer was less stable than on a negatively charged surface. *A. seyal* layer presented generally less stable behavior than *A. senegal*, regardless of the charged surface, but with a lower impact of the titanium dioxide charge surface. Moreover, the stability of the *A. seyal* film was quite similar for all surfaces whatever pH, while hydrophilic surface films seemed to have a higher stability with the salt concentration.

Surprisingly, no direct impact of surface roughness was observed between the three surfaces. Even if the adsorption was higher on the surface with the greater roughness (i.e., gold), the smoother surface presented similar adsorption capacities. However, by considering the roughness factor *r* (Table 1), we observed no variation between the three types of sensors. The surfaces were too smooth and flat (*r* close to one) to observe a significant impact of their roughness on the adsorption in the liquid state. This observation showed that roughness is not the main parameter for the adsorption capacity in the solution but should not be neglected. It is likely that the surface roughness facilitated the adsorption, and that, on similar surface properties, the rougher surface presents a higher adsorption capacity. This effect has been shown by Molino et al. [67] on BSA and fibronectin adsorption on polymeric interfaces. Moreover, the roughness impact was previously observed during the drying process of the surface after adsorption of *Acacia* gum [18].

### 3.3. Viscoelastic Properties and Conformational Change of Acacia Gum Layers—D-f Plots

To further analyze the adsorption process and gain insight into the nature of the adsorbed gums layers on the respective surfaces, $\Delta D$ vs. $\Delta F$ curves were plotted (called the *D-f* plots). As time was removed from the data, the slopes of the *D-f* plots ($|\Delta D/\Delta F|$) reflected the dynamic changes in the adsorbed layers conformation [72,73]. A high slope indicated the formation of a viscoelastic, hydrated layer, while a lower slope indicated the formation of a more rigid, less hydrated, and less viscoelastic layer. Figure 3 shows the *D-f* plots for *A. senegal* layers on the three studied surfaces in function of the ionic strength (a–c) or pH (d–e), for the overtones presenting the best signal/noise ratio (ninth), allowing a better quality of the fits but also an easy distinction of the curves on the graph. As observed, the layers presented different adsorption behaviors according to experimental conditions and surface properties. On the gold surface, the layers seemed to stiffen while the ionic strength increased. This result was directly correlated with a decrease in the hydration state of the gum layer while the ionic strength increased. While the increase in ionic strength did not present a significant impact on the surface hydrophobicity [64], it was questionable whether the decrease in the hydration state

induced by the ionic strength could increase the polarity of AGPs. Interestingly, at pH 5.0, *A. senegal* presented less viscoelastic layers, which could be related to the conformational constraint occurring on *Acacia* gum at this pH. The conformational impact was less obvious on the two other surfaces. However, the impact of the surface positive charge of $TiO_2$ on the layers' conformation was noticeable with a stiffer layer at 0 mM NaCl (b) and pH 3.0 (e). Indeed, the $TiO_2$ surface presented the highest positive charge in these experimental conditions, resulting in more rigid and probably less hydrated layers. Moreover, the adsorption was very weak at pH 3.0 on the hydrophilic surfaces, especially on the $SiO_2$ surface, with the formation of a highly "rigid" layer compared to the gold hydrophobic surface. Indeed, the "closed" structure of the protein moiety at pH 3.0 did not facilitate the adsorption on a hydrophilic surface, while a hydrophobic surface could favor the protein moiety access thanks to hydrophobic forces, therefore, increasing the viscoelastic properties by conformational rearrangements. Therefore, *A. senegal* gum layers presented a more viscoelastic behavior on the hydrophobic surface.

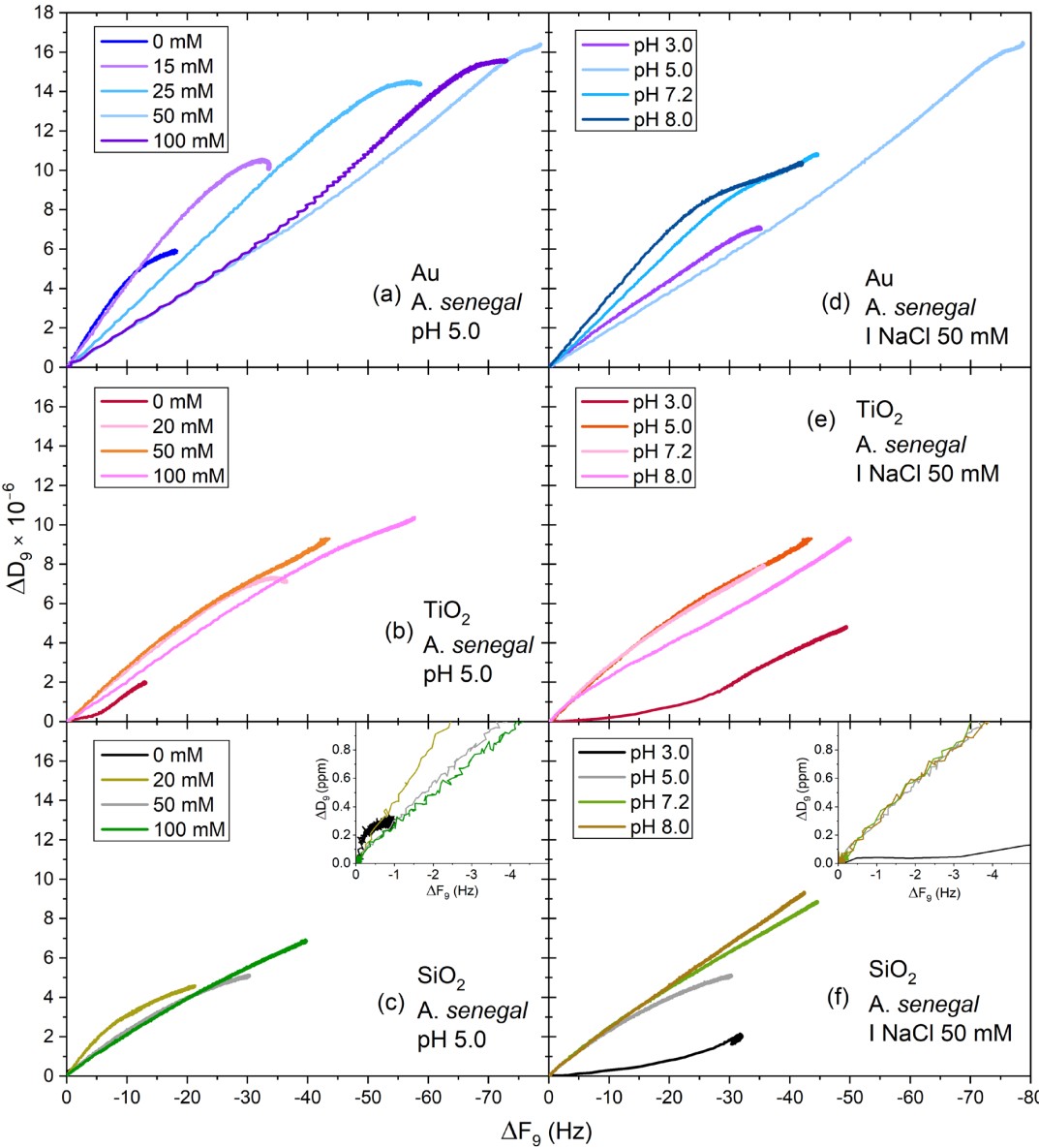

**Figure 3.** *D-f* plots from the ninth overtone of *A. senegal* in function of ionic strength at pH 5.0 (**a–c**) or in function of pH at 50 mM NaCl (**d–f**) for gold, $TiO_2$, or $SiO_2$ sensors surface during the adsorption process.

Figure S11 shows *D-f* plots for *A. seyal* layers on the three studied surfaces in function of the ionic strength (a–c) or pH (d–e). Generally, *A. senegal* layers present a more viscoelastic behavior than *A. seyal* regardless of the experimental conditions used, resulting in accordance with the previous experiments [17]. However, the *A. seyal* adsorbed layers conformational behavior was not identical. On the gold substrate, the viscoelastic behavior was opposite to the *A. senegal* layers, with an increase in the softness with ionic strength. The increase in pH increased the layers viscoelastic properties on all sensors surfaces, especially on the hydrophilic surface, in close correlation with the increase in hydration and swelling capacity of the layer.

An irregular curve with a slopes rupture was interpreted as a conformational change during the adsorption process. *D-f* plots were linearly-fitted and slope changes were related to the different regimes occurring during the adsorption process. Figure 4 illustrates the different regimes obtained for *A. senegal* at 50 mM NaCl pH 8.0 (up) and *A. seyal* at 100 mM pH 5.0 (down), as an example on gold sensors. The *D-f* plots can be divided into two to four regimes, with two to four slope changes, corresponding to the different regimes of the adsorption process. Figure S12 shows the evolution of three main regimes, displayed by the $|\Delta D/\Delta F|$ curves in function of the experimental conditions for the three types of quartz sensors. The first regime (R1) corresponded to the initial fastest step of the adsorption process, where the AGPs in their native state approached the surface. The following regimes identified by several slope changes corresponded to AGPs conformational changes to adjust its shape and maximize the interaction with the oxide surface, such as the spreading, re-orientation, and/or dehydration process (R2 to R4). A significant slope change, with a clear slope rupture, was observed with two layers (decrease in both $\Delta D$ and $\Delta F$): *A. senegal* on silica sensors at pH 3 (Figure 3f) and *A.* seyal on gold sensors at 100 mM pH 5.0 (Figure 4). These behaviors were related to a substantial change in conformation, including a mass loss of 15.0% and 9.5%, respectively, responsible for the collapse of these films.

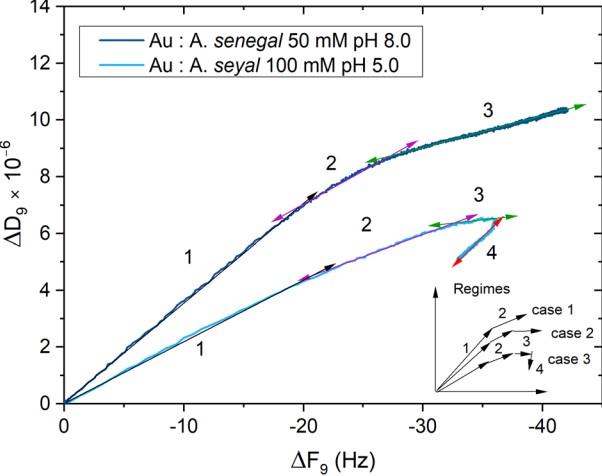

**Figure 4.** *D-f* plots slop regimes details obtained from a linear fit for *A. senegal* at 50 mM NaCl pH 8.0 (up) and *A. seyal* at 100 mM pH 5.0 (down) on gold sensors.

Generally, a stiffening of layers occurs with conformational changes, i.e., the last regime presents a more rigid behavior than the first one. These general behaviors have already been observed in previous studies [17,18], but also with protein adsorption on hydrophobic surfaces [67,74,75]. However, some exceptions should be noted: the adsorption of *A. senegal* gum on $TiO_2$ and $SiO_2$ at pH 3.0 and *A. seyal* layers on $TiO_2$ surfaces (between 20 and 100 mM NaCl), present an increase in the viscoelastic properties with the conformational change. In these cases, the adsorbed layers present an initial regime with a very "rigid"-like behavior and a low stability (strong desorption). Some authors have suggested that this behavior could be issued from a multilayer formation [72,73]. In unfavorable adsorption experimental conditions, *Acacia* gums' conformational changes reach the steadiest state

on the surface, i.e., increasing its viscoelastic properties, while, on favorable adsorption situations, gums layers would tend to more rigid conformations. Indeed, the adsorption on hydrophobic surfaces is energetically favored with the entropy gains by the loss of water from the surface and the layer, with associated conformational changes and dehydration processes, allowing for an increase in the adsorption capacity compared to the hydrophilic surface [73,76]. Therefore, *Acacia* gums present a remarkable capacity to adapt to their environment in order to reach the steadiest and most efficient state on a solid interface.

### 3.4. Interfacial Rheological Behavior of Acacia Gum Layers

Rheology in bulk is usually performed at low-frequency oscillations while QCM-D works at higher frequency with a nominal resonance frequency of 5 MHz. The instrument inertia of a QCM-D is much smaller than a classical rheometer. This allows QCM-D to be more sensitive to film properties, as microscopic relaxation processes in film occur at very short time scales [30]. Therefore, viscoelastic properties obtained by QCM-D cannot be easily compared with classical bulk data rheology. Viscoelastic properties were obtained by the QCM-D method on various soft materials [33,77–81]. Recently, Hodges et al. [82] studied Newtonian (sucrose) and viscoelastic (NaPSS) fluids with QCM-D using the Kanazawa and Gordon model, and compared the results with classical rheological methods. They found that the viscosity did not change much compared to the low frequency rheometer, while a several order of magnitude higher difference in the shear modulus was observed. However, the viscoelastic properties from QCM-D data were strongly dependent on the applied model for calculations and assumptions made.

The shear elastic modulus $\mu$ (kPa) and shear viscosity $\eta$ ($\mu$Pa·s) of the adsorbed layers were calculated from the QCM-D data using two different viscoelastic models (see Experimental section), and the results for both the gums in function of the ionic strength and pH are presented in Figure S13. The Voigt model is classically used in QCM-D analysis [29,30,34]. This model assumes the storage modulus $G\prime$ to be frequency independent, while the loss modulus $G''$ is supposed to scale linearly with the frequency [29]. As pointed out by Reviakine et al. [30], these assumptions represent an artificial limitation of the parameter space and are not usually justified. A more realistic model is a power-law model, which assume $G'$ and $G''$ to depend on frequency and is equivalent to the time-honored acoustic multilayer formalism [32]. This model was successfully applied on *A. senegal* data while the Voigt model was used for *A. seyal* films. Viscoelastic behavior cannot therefore be directly compared between the two gums.

From a general point of view, both $\mu$ and $\eta$ of *A. senegal* films increase with ionic strength on the three sensors, which is consistent with the stiffening observed with the *D-f* plot analysis while both moduli seem to be less influenced with increasing pH. The lower pH, however, significantly changes the layer modulus, with an increase of shear viscosity for TiO$_2$ surface while both moduli increase on silica surface. This is correlated with the more "rigid" behavior of these layers observed in Figure 3. According to Voigt modelling, *A. seyal* layers moduli decrease between 0 and 20 mM NaCl in solution, due to the softening of layers (see *D-f* plot in Figure S11), and then slowly increase in function of ionic strength while a general decrease is observed with the increase in pH. The comparison between the power-law and Voigt model for *A. seyal* adsorption on gold substrate is displayed in Figure S13. The shear viscosity moduli are quite similar between both models, while the shear elastic modulus is much lower with power-law modelling. The shear viscosity moduli are, therefore, less impacted whatever the viscoelastic model used, contrary to the shear elastic moduli. According to these results, the *D-f* plot can be sufficient to describe the general viscoelastic behavior of layers, without the use of a model, and thus compare all experimental data.

An attempt to obtain further insight of the gums rheological properties was carried out by calculating the loss tangent, tan $\delta$ (Figure 5), according to the following equation:

$$\tan \delta = \left( \frac{G''}{G\prime} \right) = \left( \frac{2\pi f \eta}{\mu} \right) = \left( \frac{\omega \eta}{\mu} \right) \tag{4}$$

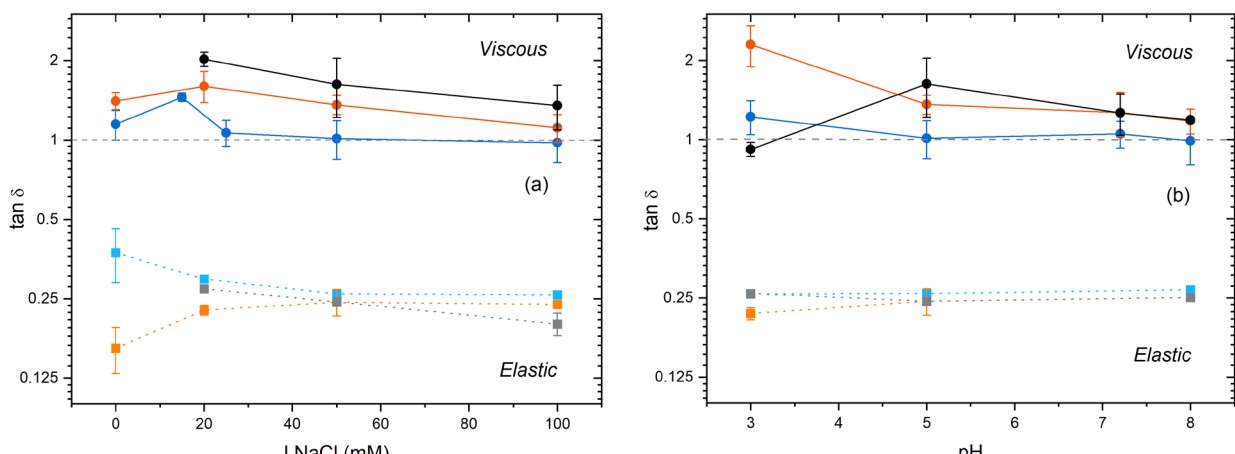

**Figure 5.** Evolution of tan $\delta$ in semi-log scale as a function of ionic strength at pH 5.0 (**a**) or in function of pH with 50 mM NaCl (**b**) for *A. senegal* with the power-law model (-•-) and *A. seyal* with the Voigt model (··■··) onto gold (blue), TiO$_2$ (orange), or SiO$_2$ (black) sensors surface. *Dash line is the transition between an elastic and a viscous behavior.*

The layers exhibited mainly elastic behavior if tan $\delta < 1$ and predominantly viscous behavior if tan $\delta > 1$. If tan $\delta = 1$, meaning $G\prime = G''$, the layers presented a viscoelastic behavior. An increase in tan $\delta$ values indicated that the layer was more able to absorb more energy without elastically returning it, acting more as a shock absorber [83]. Interestingly, the results strongly depended on the viscoelastic model used. For *A. senegal*, the power-law model showed that the layers presented mainly viscous behavior on silica and titanium dioxide, i.e., hydrophilic surfaces, while the gold substrate gave viscoelastic layers, i.e., hydrophobic surface. These results were quite surprising because it is well known that *Acacia* gum layers present an elastic behavior on interfacial emulsion [84–87]. However, the time-dependent rheological properties were previously observed in gum dispersion at 6 wt% with the evolution from a structured liquid viscoelastic to solid-like viscoelastic properties after ageing [88]. Since the QCM-D measurements were performed within a few hours at a very low concentration, and considering that the *A. senegal* layers were highly hydrated (>90%) [17], a general liquid-like behavior made sense on the solid surface.

According to the Voigt model, *A. seyal* layers would present elastic behavior on all sensors surface (Figure 5), with the results consistent with the more "rigid" behavior observed with the *D-f* plots. However, the data obtained with the power-law model for *A. seyal* on gold substrate showed completely different results (Figure S14), with viscous behavior for all experimental conditions. The rheological parameters obtained from QCM-D data modelling should be considered with care and strongly depend on the model used.

Elmanan et al. [89] compared the rheological properties of both *A. senegal* and *A. seyal* gums at the air/liquid and liquid/liquid interfaces. They found that *A. seyal* gum presented a viscous surface compared to the elastic surface observed with *A. senegal* gum. These discrepancies could be explained by several factors. Firstly, as discussed in the first part of this section, the QCM-D rheological measurement did not record the same relaxation process as a classical rheometer. In addition, the concentration of gum used in this study was much lower than that used in classical rheology (150 or 500 ppm compared to ≥10,000 ppm minimum). Additionally, the QCM-D measured the layer on a solid surface while previous rheological studies were performed on solid-liquid (dispersion) or liquid-liquid (emulsion) interfaces. The QCM-D measurement highlighted the differences in the behavior of the rheological properties that occurred at very low concentrations on solid surface at a short time scale.

## 4. Conclusions

The present study highlights that hydrophobic forces have higher impact than electrostatic forces on *Acacia* gum adsorption on the oxide surface. The *Acacia* gum adsorption capacity is higher on hydrophobic surfaces than on hydrophilic surface. However, the stability of the layers increased on a negatively charged surface. The initial structural conformation of AGPs in solutions, and its net charge, is a crucial parameter for the adsorption process, especially at high pH levels where the degree of hydrophobicity of the surface has a lower impact.

The viscoelastic properties of adsorbed layers were evaluated for both *A. senegal* and *A. seyal* gums. As previously observed, the *A. senegal* gums layers present a high viscoelastic behavior, especially on hydrophobic surfaces, compared to the *A. seyal* layers. The *D-f* plots show that various rearrangements, conformational changes, or dehydration process occur after the first step of adsorption, depending on the experimental conditions, in order to stabilize the layer. This result shows the remarkable capacity of *Acacia* gum to adapt to the surface properties in order to reach the steadiest state on the solid surface.

Attempts of rheological analysis with QCM-D method were carried out, and the results showed that the *A. senegal* layers presented a viscous behavior on the hydrophilic surface and viscoelastic behavior on the hydrophobic surface. *A. seyal* layers showed elastic behavior whatever the surface used according to the Voigt Model or a viscous behavior on the hydrophobic surface when considering the power-law model.

A summary of the main conclusions on *Acacia* gum adsorption highlighted by the present study is presented in Figure 6.

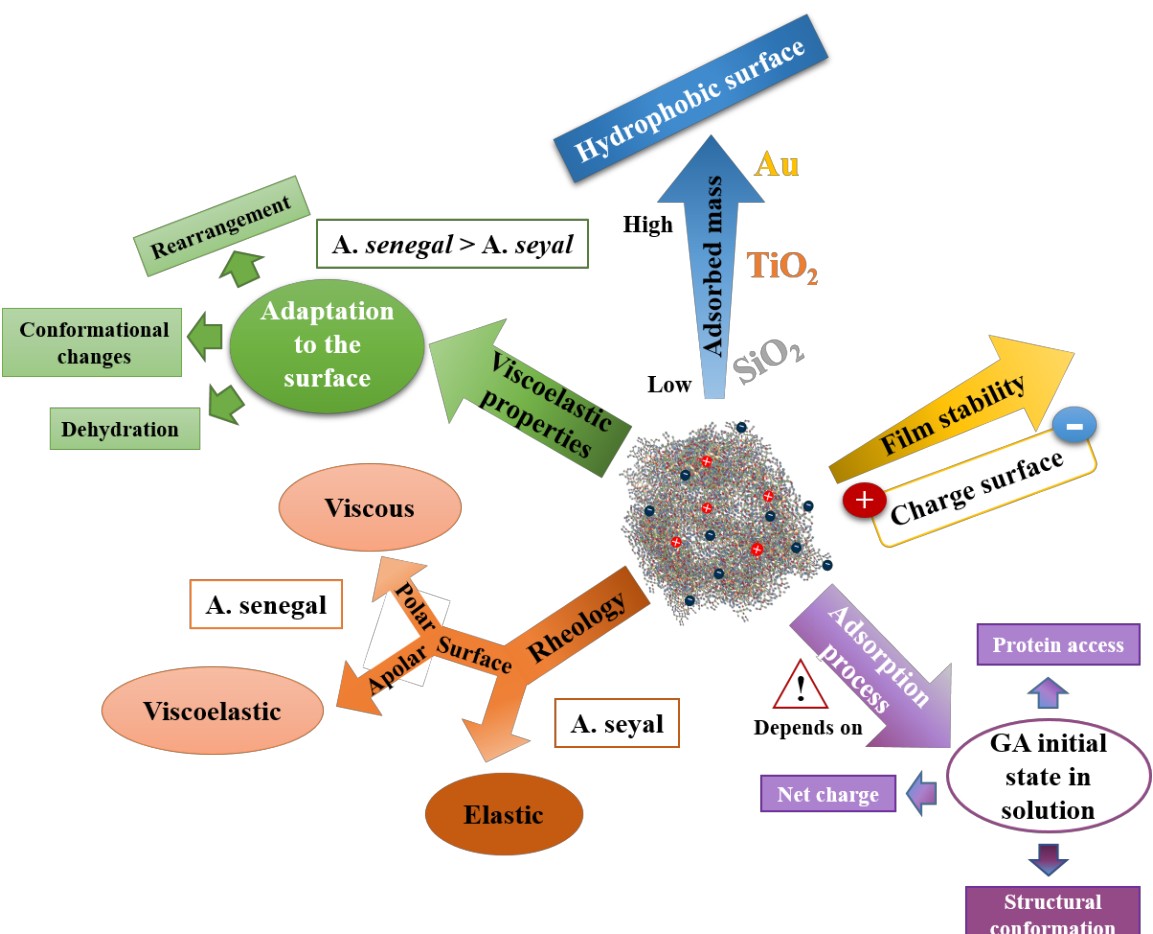

**Figure 6.** Summary of the main conclusions of *Acacia* gum adsorption onto solid surfaces.

**Supplementary Materials:** The following supporting information can be downloaded at: https://www.mdpi.com/article/10.3390/colloids7020026/s1, Table S1. Biochemical composition and structural parameters of *A. senegal* and *A. seyal* gum; Figure S1. AFM topologic images of quartz sensor surface; Figures S2–S8. QCM-D raw data; Figure S9. QCM-D results for *A. seyal* for the two viscoelastic models; Tables S2–S13. QCM-D calculated results; Figure S10 Films desorption in function of salt concentration and pH; Figure S11. *D-f* plot for *A. seyal* adsorbed films; Figure S12. |$\Delta D/\Delta F$| *D-f* slopes obtained from a linear fit of the three principal regimes in function of salt concentration and pH; Figure S13. Shear elastic modulus μ (kPa) and shear viscosity η (μPa·s) in function of salt concentration and pH; Figure S14. Evolution of tan δ (G″/G′) for *A. seyal* for the two viscoelastic models.

**Author Contributions:** A.D. contributed by designing, performing the experiments, analyzing data, and writing the paper. M.N., C.S. and D.R. contributed by analyzing data and writing the paper. All authors have read and agreed to the published version of the manuscript.

**Funding:** This work was financially supported by the Alland and Robert Company.

**Data Availability Statement:** Data can be available upon request from the corresponding author.

**Conflicts of Interest:** The authors declare no conflict of interest.

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
