# Peer review of "Impact of Hydrophobic and Electrostatic Forces on the Adsorption of Acacia Gum on Oxide Surfaces Revealed by QCM-D"

_colloids, doi:10.3390/colloids7020026_

Round 1

Reviewer 1 Report

In this paper, the authors studied the adsorption behaviors of two types of Acacia gums on different oxide surface and analyzed the impact of surface properties on their adsorption behaviors. The paper needs major revision before the publication. Details are as follows:

1.                  In this paper, the interaction between gum and oxide surface was studied. In most cases, only surface charge and hydrophilicity were taken into consideration. However, other forces, such as coordination and gold-thiol interaction, were ignored. Why? Further, to analyze the influence of surface properties, the authors can use modified SiO2 surface with different hydrophilicity and zeta-potential. However, the authors used surfaces with even different chemical properties. Why?

2.                  What was the essence of hydrophilic interaction herein?

3.                  In this paper, the authors cited Ma’s work which suggested that Acacia gum would present an expanded structure at pH 8.0 due to the increased repulsion of charged amino acids at high pH conditions. Owing to the nature of amino acids, an expanded structure may be also found at very low pH. As a result, what was the range of pH for stable gum structure? For the change of gum structure, DLS test is suggested to be done.

4.                  In Figure 3, two special curves (0 mM in b and pH 3.0 in e) could be found. The slope of these two curves gradually increase with the increase in ΔFg, which was different from the trends illustrated in Figure 4. What was the reason for that?

5.                  In this work , two type of gums, A. senegal and A. seyal gums, were used for studying their adsorption behaviors on varied oxide surfaces. Their different adsorption behaviors were shown in the manuscript. However, the influence of their structural similarity and dissimilarity on the adsorption was not clearly illustrated.

Reviewer 2 Report

This manuscript presents an intensive experimental study of the adsorption of Acacia gums extracted from two different Acacia trees. The study is done at different pHs and ionic strengths, using NaCl salt. The adsorption of the gums has been done using different solid surfaces that show different contact angles with waer. The data are analyzed using commercial software. Since the Acacia gums are used in many industrial applications, the study is timely. The experiments are well performed, using a commercial technique. The manuscript fits within the priorities of the journal. However, some revisions are necessary before it can be accepted for publications.

COMMENTS: Please see the attachment.

Round 2

Reviewer 1 Report

The authors have addressed all my concerns. It can be published as it.

Reviewer 2 Report

The authors have performed an extensive revision of the original manuscript, and have taken into account my comments on it. Therefore, I consider that it can be accepted for publication in its present form.